# Towards Effective Environmental Sustainability Reporting in the Large Industrial Sector of Bahrain

**Abdulkarim Hasan Rashed** [1,*], **Suad Ahmed Rashdan** [2] and **Ahmed Y. Ali-Mohamed** [2]

1   Environment and Sustainable Development Program, College of Science, University of Bahrain, Sakhir 32038, Bahrain
2   Department of Chemistry, College of Science, University of Bahrain, Sakhir 32038, Bahrain; suadr@hotmail.com (S.A.R.); ahmedyobh@gmail.com (A.Y.A.-M.)
*   Correspondence: kme2001@hotmail.com

**Abstract:** The industrial sector plays a vital role in economic development; therefore, there is a necessity to integrate sustainability into industrial development to maintain the economy and avoid any degradation impacts on the environment, and thereafter on society. Thus, do Bahraini companies have sustainability reports and if so, are these reports based on GRI guidelines? Has the status of their sustainability reports been analyzed? This research aims to examine the sustainability reports of companies by analyzing the status of sustainability aspects in their materiality matrices to assist in identifying and prioritizing the most significant sustainability issues for advancement in their future reporting and to improve their environmental performance. This study employs a content analysis approach and analyzes 11 reports from the period 2016–2020 for three companies in Bahrain's large industrial sector. The study reveals that the companies using materiality analysis in their reporting benefit from better monitoring and measuring of their environmental performance, and from implementing SDGs. Furthermore, the study indicates that the utilization of a materiality matrix as a reporting tool can define and improve report contents by considering stakeholders' views, consequently, improving the quality of the sustainability reports. The study concludes by proposing a set of recommendations.

**Keywords:** sustainability matrix; reporting tools; GRI; materiality analysis; content analysis



## 1. Introduction

The industrial sector is a major part of the business sector, and it is more inclined to have negative sustainability effects due to its direct or indirect impact on the environment. This raises concerns about attaining environmental sustainability for industrial-based development; therefore, it is essential to develop strategies based on an industrial perspective for enhancing the performance and implementation of the sharing economy [1–3]. Accordingly, conflicts have arisen between the economic consequences that have led to industrialization and the dangerous negative impacts of the devastation and consumption of natural resources and severe pollution [4,5]. In this regard, Kopnina stated that when economic gains are placed above environmental protection, economic development goals are likely to outweigh environmental concerns [6]. Barbier pointed out that there is agreement that ecosystems should be considered as economic assets, as they provide the economy with necessary goods and services [7]. In the same context, Dahlmann et al. showed that the "purpose ecosystem" consists of various actors seeking to broadly improve private sector aims and encourage the sector to shift from seeing sustainability as a side issue, being part of "business as usual" and profitability, to focusing on including sustainability considerations in their business strategies [8].

Linking industry with the environment is an essential part of industrial enterprise performance [9]; thus, the industrial sector is an important area for the development of sustainability towards a better future [10,11], and it is imperative to observe the related

transitions in this sector from the perspective of sustainability science [12]. The industrial trend towards sustainable development clearly occurs through an improvement in the production operations that rely on the technologies that utilize resources more efficiently, generate minimal waste [13], and recognize air emissions as parameters for environmental sustainability [9]. In addition, Oliveira et al. reported that, in the early 1990s, the concerned companies focused on environmental impacts resulting from their production operations by applying the end-of-pipe treatment principle and recycling practices as a preventive and proactive process [14]. Since that time, sustainability assessment has increasingly grown as a significant new evaluation approach for assisting both decision-making and policy-making on a wide range of sustainability components in various sectors globally [15], such as energy, transport, industry, infrastructure, finance, and land. In addition, more consideration has been given to the industrial sector and, specifically, the most carbon-intensive industries [16].

Industrialization is a dynamic economic process that creates income and employment, enhances businesses, and uses resources efficiently [17]. The United Nations 2030 Agenda for Sustainable Development promotes "sustainable industrialization" under Goal 9, which means "building resilient infrastructure, promote inclusive and sustainable industrialization and foster innovation" [18]. Industrial development is considered to be a key driver of economic growth because the opportunities for major economies and technology are multiplied [1]. The sustainable manufacturing notion emanates from sustainable development concepts [19,20], and is one of the environmental initiatives [21] and an essential notion that must be adopted to reach full sustainable development [22]. This can be achieved through production operations with minimal environmental impact by employing the product lifecycle concept and environmentally integrated strategies within the management of manufacturing processes [19]. In addition, globally, several industries have recognized the significant economic and environmental benefits of sustainable manufacturing practices [23]. For instance, progressing concerns about environmental issues from greenhouse gas emissions, natural resource shortages, toxic waste production, and rising stresses on sustainable and clean energy has led to the development of a green manufacturing paradigm to minimize the degradation and harmful environmental impacts affecting the planet [24].

In the 1970s and 1980s, environmental legislation became more robust by requesting companies to prove their compliance; hence, the environmental and social impacts of companies have become increasingly important [25]. Consequently, the concept of sustainability compliance has provided further opportunities for research [26]. Different features of the industrial sector influence the reporting practices concerning sustainable development goals (SDGs), though the existing monitoring and management practices of the SDGs are still far from achieving an optimality level both nationally and globally [27,28]. For instance, Bakardjieva claimed that the inability of most companies to attain SDGs is due to a failure to employ sustainable practices, e.g., remanufacturing, reusing, and recycling [29]. Jiang et al. concluded that there is a large gap in the sustainability reporting of companies, and the majority of the reports recommended taking actions to improve sustainability performance [2]. Therefore, measuring, assessing, and developing an industrial sustainability performance is a critical issue [30–32]. Usually, developing countries faces problems when implementing new strategies, particularly with sustainable benefits, due to a lack of regulations, awareness, and knowledge [3].

Based on the survey of the literature that follows, this article reviews sustainability reporting and its tools, along with SDG reporting, and examines the relationship between the materiality matrix and sustainability reporting. Moreover, it focuses on the industrial sector in Bahrain and its sustainable reporting. The methodology employed in reviewing the sustainability reports of the Bahrain industrial sector, a discussion of the results, and a conclusion with several recommendations are included in this article.

This study contributes to the literature and, -to the best of our knowledge- is the first to examine whether Bahraini companies have sustainability reports. If so, are these reports

based on GRI guidelines? Has the status of their sustainability reports been analyzed? Aimed at attaining a deep and broad understanding of the status of SDG reporting among the large industrial sector in BahrainThe result of this study and analysis will assist in identifying and prioritizing the significant issues of sustainability to advance their future reporting for better environmental performance, and to encourage other companies that did not establish sustainability reports to follow suit.

## 2. Sustainability Reporting Tools and SDG Reporting

In 1987, after the publication of the Brundtland Report (*Our Common Future*), companies reporting content focused more on particular environmental issues, such as air pollution, wastewater, and water management, with a continuing incorporation of social issues in the reports. From 2016 onward, company sustainability reports have focused on the SDGs of the UN 2030 Agenda [27]. As companies vary in their activities, they differ in their contributions to the implementation of SDGs [33]. Therefore, the content of sustainability reports differs among companies in terms of environmental aspects and reporting quality, and presents substantial sustainability issues and practices [34]. Significantly, however, SDGs can play an essential role in the promotion of sustainability reporting [35], and the complete achievement of all 17 SDGs requires companies to contribute greatly to the organizational, administrative, and reporting aspects [36]. In this regard, van Zanten and van Tulder noted that if companies can improve their ability to include all SDGs, their sustainability strategies will be more successful [33]. Further, PwC stated that the innovative characteristics of the SDGs can broaden the reporting quality concept [37]. According to Acuti et al., the starting point of action to achieve SDGs lies in more attention given to reporting the effects of companies on resilience [38]. To be sustainability efficient, Engert et al. observed that sustainability reporting should be harmonized and components incorporated into the company strategy [39]. In the same context, Haywood and Boihang highlighted that companies need to demonstrate their obligation to SDGs via reporting the alignment of their strategies, programs, actions, or initiatives with specific goals and targets, and how they integrate those goals within their business plans and strategies [40]. Furthermore, Stewart et al. reported that companies need to proactively manage environmental issues associated with their activities and incorporate them into their environmental sustainability strategies [34]. In addition, SDGs afford a framework for sustainable and resilient business growth strategy [41].

During the period 1979 to 1989, the world witnessed unmatched environmental disasters; thus, the obligation of the private sectors in the localization of sustainable development issues became highly crucial [42] (p. 193). Industrial development and urban growth are usually associated with severe pollution and resource consumption [5], which have increased environmental degradation and severely affected the quality of life [43,44]. Consequently, the formulation of appropriate corporate reporting systems that address company performance rather than their financial performance are crucial [45]. A case in point, in 1989, the Coalition for Environmentally Responsible Economies (CERES) issued a set of 10 principles, extending from common environmental concepts to specific disclosures, aimed at creating a framework for the information flow needed to assess company performance based on historical context and societal norms. Those principles represent a milestone in shaping a unified environmental reporting system that companies can voluntarily comply with, and led to the birth of the Global Reporting Initiative (GRI) in 1997, which combined the reporting standards of governance, economics, and corporate social responsibility (CSR) into a unified sustainability reporting framework for the global marketplace [46]. As such, the GRI Standards guidelines have become a voluntary reporting reference widely adopted by companies globally [34,37,47,48]. The first effort to incorporate SDGs through nonfinancial indicators was developed via a collaborative initiative among the United Nations, the Global Reporting Initiative (GRI), and the World Business Council for Sustainable Development (WBCSD), providing the SDG Compass to support companies in tailoring strategies, and in managing and measuring their contributions to SDGs [49]. Recently, the European Union

displaced the misguided "nonfinancial" concept with "sustainability" by issuing a new proposed Directive on Corporate Sustainability Reporting that could involve around 50,000 European companies [50]. Historically, the evolution of corporate sustainability can be classified into three essential periods: the first period (1950–1990) marks the era of corporate social responsibility (CSR); the second period (1990–2005) presents a noteworthy evolution of CSR and the emergence of the triple bottom line (TBL) concept (economic prosperity, environmental quality, and social equity); and the third period (2005–2020) describes the development of modern corporate sustainability as a result of several novel concepts that have become commonplace, such as ESG (environmental, social, and governance) [51] (p. 66). Table 1 summarizes the sustainability reporting tools.

**Table 1.** Sustainability reporting tools: frameworks, indices, and standards.

| # | Framework/Index/Standard | Developer/Founder | Year Established | Reporting Sectors | Cost | Key Areas of Reporting | Related SDGs | References |
|---|---|---|---|---|---|---|---|---|
| 1 | SIGMA project | Motorola by Bill Smith | 1986 | All companies | Yes | Quality improvement, reduce waste, improve processes, and increase customer satisfaction | 9, 12, 13 | [52] |
| 2 | ISO 9001 standard | International Organization for Standardization (ISO) | 1987 | All companies | Yes | Quality management system | 1–16 | [53–55] |
| 3 | DPSIR framework | Organization for Economic Co-operation and Development (OECD) | Late 1990s | All companies | No | D: driving forces (economic sectors, human activities) P: pressures (emissions, waste) S: states (physical, chemical, and biological) I: impacts on (ecosystems, human health, and functions) R: responses (prioritization, target setting, and indicators) | All | [56,57] |
| 4 | Greenhouse Gas Protocol (GHG Protocol) | World Resources Institute (WRI) and the World Business Council for Sustainable Development (WBCSD) | Late 1990s | Large companies | No | Greenhouse gas (GHG) emissions | 13 | [58,59] |
| 5 | Eco Management and Audit Scheme (EMAS) | European Commission | 1993 | All companies | Yes | Environmental management system | 6, 7, 11, 12, 13, 14, and 15 | [60] |
| 6 | Triple Bottom Line (TBL) | John Elkington | 1994 | All companies | No | Economic prosperity, environmental quality, and social equity | All | [61] |
| 7 | World Business Council for Sustainable Development (WBCSD) | CEO-led organization | 1995 | All companies | No | Circular economy, cities and mobility, climate and energy, food and nature, people and society | All | [62,63] |
| 8 | ISO 14001 standard | International Organization for Standardization (ISO) | 1996 | All companies | Yes | Environmental management system | 6, 7, 8, 9, 12, 13, 14, and 15 | [53,55,64,65] |

**Table 1.** *Cont.*

| # | Framework/Index/Standard | Developer/Founder | Year Established | Reporting Sectors | Cost | Key Areas of Reporting | Related SDGs | References |
|---|---|---|---|---|---|---|---|---|
| 9 | Global Reporting Initiative (GRI) | Global Sustainability Standards Board (GSSB) | 1997 | All companies | No | Environment, social, and governance | All | [66] |
| 10 | The UN Global Compact | Member of the United Nations Sustainable Stock Exchanges (SSE) initiative | 2000 | All companies | No | Stating 10 principles in the areas of human rights, labor, environment, and anticorruption | All | [67] |
| 11 | Carbon Disclosure Project (CDP) | Founder Paul Dickinson | 2000 | Large companies, cities, countries, regions | No | Environment and governance | 6, 7, 11.b, 12.2, 12.6- 12.8, 12.a, 13.1–13.3, 13.a, 15.1- 15.3, 15.5, 15.7, 15.b | [68,69] |
| 12 | Competitive Industrial Performance (CIP) index | UNIDO | 2003 | All companies | No | • Manufacturing value added (MVA) per capita<br>• Manufactured exports per capita<br>• Industrialization intensity<br>• Export quality | 9 | [70] |
| 13 | ISO 4040 standard | International Organization for Standardization (ISO) | 2009 | All companies | Yes | • Lifecycle assessment | 9 | [71] |
| 14 | The International Integrated Reporting Council (IIRC) | Founders Robert Eccles and Mike Krzus | 2010 | All companies | | Environment, social, and governance | All | [72,73] |
| 15 | ISO 26000 standard | International Organization for Standardization (ISO) | 2010 | All companies | Yes | Guidance on social responsibility | All | [55,64,74] |
| 16 | The Sustainability Accounting Standards Board (SASB) | Founder Jean Rogers | 2012 | Large companies | Yes | Environment, social, and governance | All | [69,75] |

**Table 1.** *Cont.*

| # | Framework/Index/Standard | Developer/Founder | Year Established | Reporting Sectors | Cost | Key Areas of Reporting | Related SDGs | References |
|---|---|---|---|---|---|---|---|---|
| 17 | Green Industrial Performance (GIP) index | UNIDO | 2014 | All companies | No | • Capacity to produce and export green manufacturers<br>• The role of green manufacturers<br>• Social and environmental aspects of green manufacturers | 9 | [76–78] |
| 18 | SDG Industry Matrix: Industrial Manufacturing | United Nations Global Compact and KPMG | 2016 | All companies | No | • Opportunities for shared value<br>• Good practice principles, standards, and tools<br>• Multiple stakeholder partnerships and collaborations | All | [79,80] |
| 19 | Task Force on Climate-related Financial Disclosures (TCFD) | Financial Stability Board (FSB) | 2017 | All companies | No | Environment and governance | 7,9,12,13 | [69,81] |
| 20 | ISO 45001 | International Organization for Standardization (ISO) | 2018 | All companies | Yes | Replaced BS OHSAS 18001 (the Occupational Health and Safety Assessment Specification) standard | 3,5,8,9,10,11,16 | [53,55,82] |
| 21 | SDG Impact Standards for Enterprises | UNDP | 2020 | All companies | No | Strategy, management approach, transparency, and governance | All | [83] |
| 22 | SDG Impact Assessment Tool | Gothenburg Centre for Sustainable Development (GMV) | 2020 | All companies | No | Assessing the SDGs for direct and indirect impacts | All | [58,84] |

Sustainability reporting is a significant driver of, and catalyst for, the sustainability of companies [85,86], including environmental and social aspects [87], and acts as an incentive to integrate SDGs within the business model [35,88]. In addition, sustainability reporting assists companies to understand, measure, communicate, and lead efforts towards setting SDGs, clarifying internal aims, and leading the shift towards further sustainable development [89]. Adding to that, sustainability reporting and its disclosure tools are the key methods for external communication with stakeholders [39] to keep, gain, or improve the reputation of the company [90], and to satisfy the stakeholders' requirements for greater transparency on sustainability issues [85]. Furthermore, presentation of the information and transparency are the basic principles of the quality of a report [91]. Corporate sustainability reporting is defined as "a robust incentive and strategic tool for a company to disclose their contribution practices to the SDGs, and assists in engaging the stakeholders, supporting the sustainable decision-making process on all levels, forming the business strategy, leading innovations, best performances, better value creation, and brings investments" [92,93]. A recent study by García-Sánchez et al. found that environmental performance plays a mediating factor in the positive relationship between environmental innovation and integrated environmental disclosure [94]. Another study conducted by Gerged et al. in the Gulf Cooperation Council countries (GCC) region reported that corporate environmental disclosure is significantly and positively related to a company's value [95]. Thus, sustainability reporting is the disclosure of both the impacts and the progress of a company's environmental, economic, and social policies in the context of SDGs.

SDGs are significant drivers for sustainability in many fields, including the industrial sector and the green economy [11,96]. There are some factors that may affect the quality of the disclosure on SDGs, such as the listing status, industry size and type, certification of corporate social responsibility (CSR), external confirmation of sustainability reporting [27], environmental sensitivity, corporate governance, ownership structure, stakeholder pressure, environmental performance, board characteristics [94], and board size and independence [97]. The industrial sector is considered to encompass factors that can influence the reporting practices that companies adopt concerning SDGs, because of the various characteristics of each industrial sector and their level of impact on sustainability dimensions [27]. Currently, many companies have started an investigation into SDGs [98], and there has been an increase in the number of international companies disclosing their environmental and social performances in their annual reports [99]. Empirically, it has been reported that very few companies mention SDGs in their reports [98,100], but a high level of SDG reporting can be found in the larger companies [25,47,100]. For instance, a study conducted by KPMG revealed that only 40% of the top world companies documented SDGs in their annual reports, and of these, 84% identified the most relevant SDGs to their business [101]. Although SDGs are important to the industrial sector, there are few studies on this subject, thus there is a need to study this sector in different countries, especially on a national scale [11]. The reporting of SDGs is still under investigation [35], and existing monitoring and management of SDGs at both national and global levels are still far from being at an optimal level. Furthermore, there is a dearth of studies on the interactions and interlinkages among SDG indicators for the industrial development and protection of the environment [5,28]. However, few companies know how to implement SDGs in their practices and report on their achievements [102].

## 3. Sustainability Reporting and Materiality Matrix

In 2013, the GRI launched the G4 guidelines, which present a materiality matrix (see Figure 1) for the identification of the related topics between a company and their related stakeholders. Thus, a report should include concerned stakeholders who are either greatly influenced by the company or who themselves influence the successful implementation of the company objectives and address the related topics [103]. The materiality matrix is the most utilized method to define materiality issues [32,104], and the most important and complex among other standards [105] at the global level [106]. The concept of materiality

is understood as a reporting threshold [107,108], and is the main guideline to reducing the issue of low credibility [109]. In describing materiality, the GRI clarifies that "the report should cover aspects that: reflect the organization's significant economic, environmental and social impacts; or substantively influence the assessments and decisions of stakeholders" [110]. The materiality matrix can also be defined as "a typical tool to report on compliance and dissent of stakeholder and management views in sustainability and integrated reporting" [111]. Further, GRI G4 guidelines focus on a multiple stakeholder approach rather than sustainability reporting standards, and they provide the flexibility to report on the issues that are most significant to the company and its related stakeholders [112]. The guidelines also encourage stakeholder participation in the materiality analysis, meaning that companies can improve their liability to their stakeholders and make sustainability efforts that are greatly effective [113]. In the views of Ortar, there is still a need for research contributions to explore the area of materiality to expand the applications of this methodology [114]. For instance, one of the research issues concerning the definition of materiality is that there is no final definition of materiality in inferential sustainability research or at an institutional level [104]. In addition, Ferrero-Ferrero et al. studied evidence showing that materiality assessment still faces many constraints in sustainability reports [32].

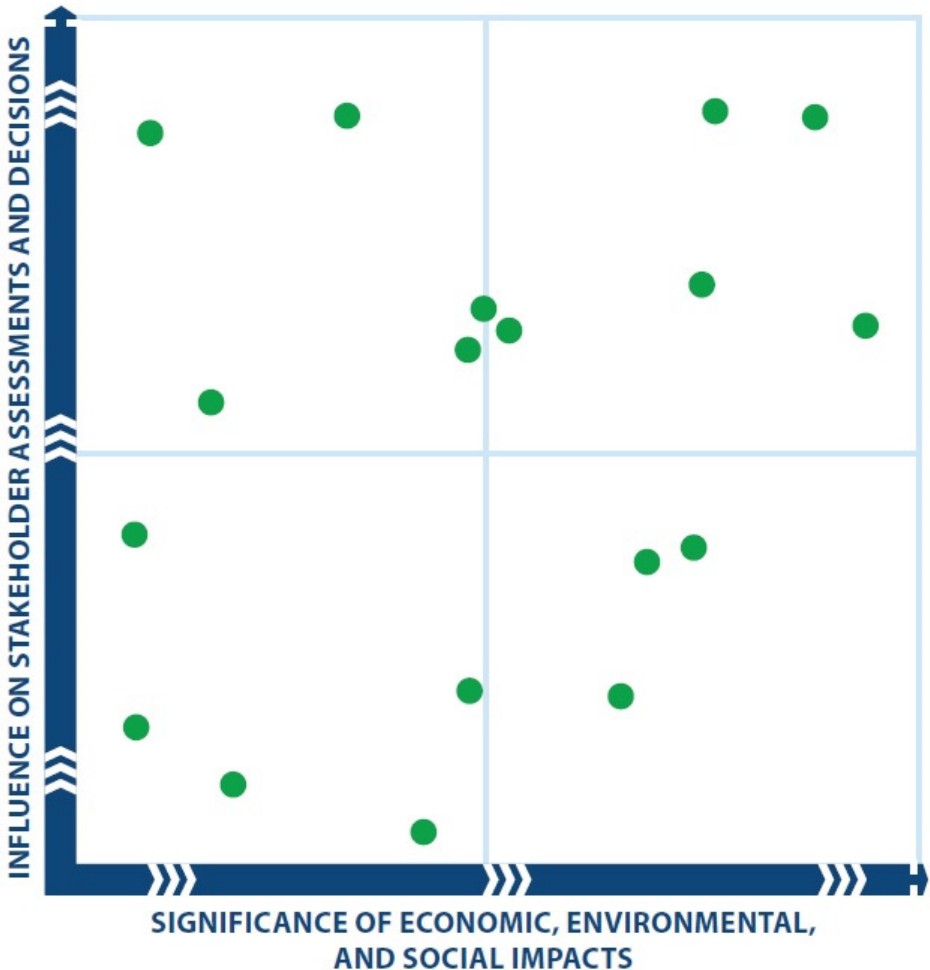

**Figure 1.** The Global Reporting Initiative (GRI) materiality matrix [107].

The materiality analysis process aims to identify and prioritize pertinent aspects and issues of the company, the result of which is a materiality matrix that designates the company and stakeholder views on the importance of related issues [115]. In the same notion, materiality analysis is perceived as a methodical and strict process that contributes

to identifying important stakeholder-oriented measures of corporate sustainability [113]. Thus, materiality analysis is essential to corporate sustainability because it provides the necessary integrated approach to setting sustainability strategies and preparing sustainability reports [116]. In this regard, some companies have admitted to identifying significant sustainability issues related to stakeholders and reflected these views in formulating strategies and corporate reporting [117]. Remarkably, there has been an increase in the number of large companies seeking to adopt the materiality approach as an integral part of their sustainability reporting [113], and a transitional phase has been observed in sustainability reporting, shifting from being voluntary to something more concrete [114]. Eventually, it is intended that the credible materiality analysis process will overcome the existing shortcomings in the authenticity and completeness of corporate sustainability reporting [118].

## 4. The Industrial Sector in Bahrain

The Ministry of Industry, Commerce, and Tourism (MICT) commands Bahrain's industrial sector, and its strategy for this sector relies on more development and diversification of the current main industries, such as the aluminum and petrochemicals sectors, and establishing other basic industrial sectors (see Figure 2) with high added value in a bid to increase the manufacturing contribution of the sector to the economy [119]. Bahrain's Economic Vision 2030 focuses on sustainability, and Bahrain has adjusted the portfolio of foreign investments inducing them to reduce the reliance on heavy industries, which depend mainly on gas to generate electricity and may negatively affect air quality [120].

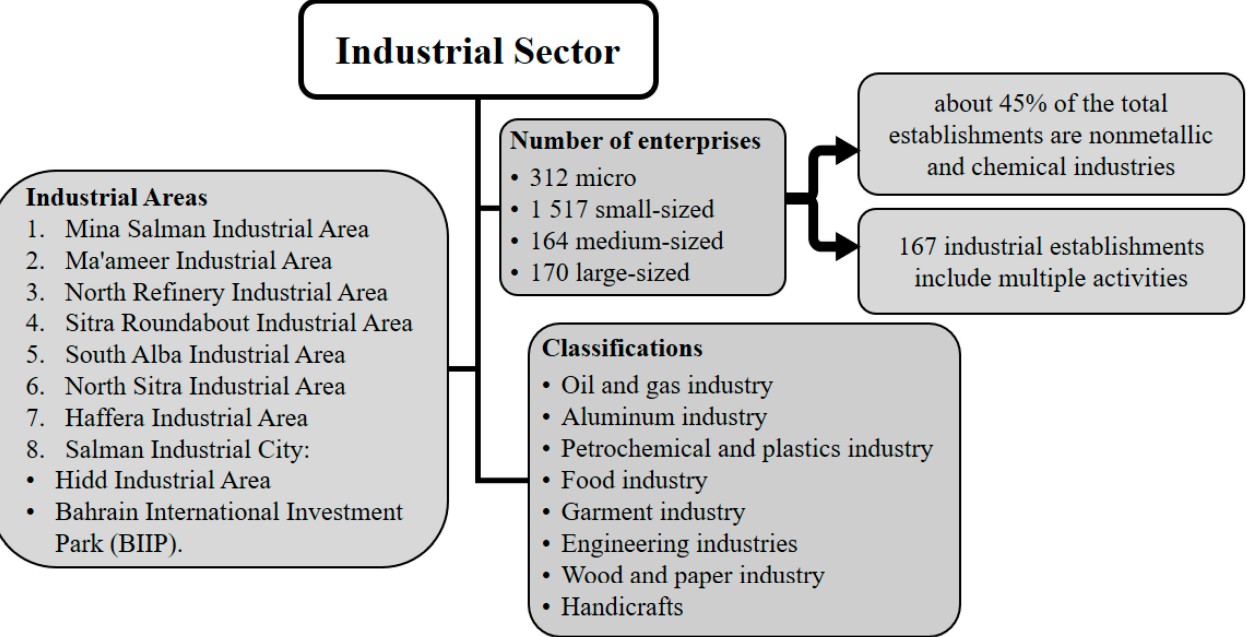

**Figure 2.** An overview of the components of Bahrain's industrial sector (information and data sources: [121,122]).

Private industries produced 42.22% of the total gross domestic product (GDP) for 2017, and the workforce in this sector was 92,857 [123]. The number of enterprises, based on the 2018 records of the MICT, increased to 2163 compared with 366 industrial establishments in 2015, i.e., the rate of increase grew by 490.9% during the three years [123]. Furthermore, the various industrial areas attracted a range of light exports and medium-sized companies comprising aluminum extrusions and assembly, paper, steel, asphalt plants, wool and wire-mesh, cables manufacturing, plastics, and marine services, with prefabricated furniture and buildings being established [119,121–124]. The major activity classifications of Bahrain's industrial sector (with percentages) are shown in Figure 3.

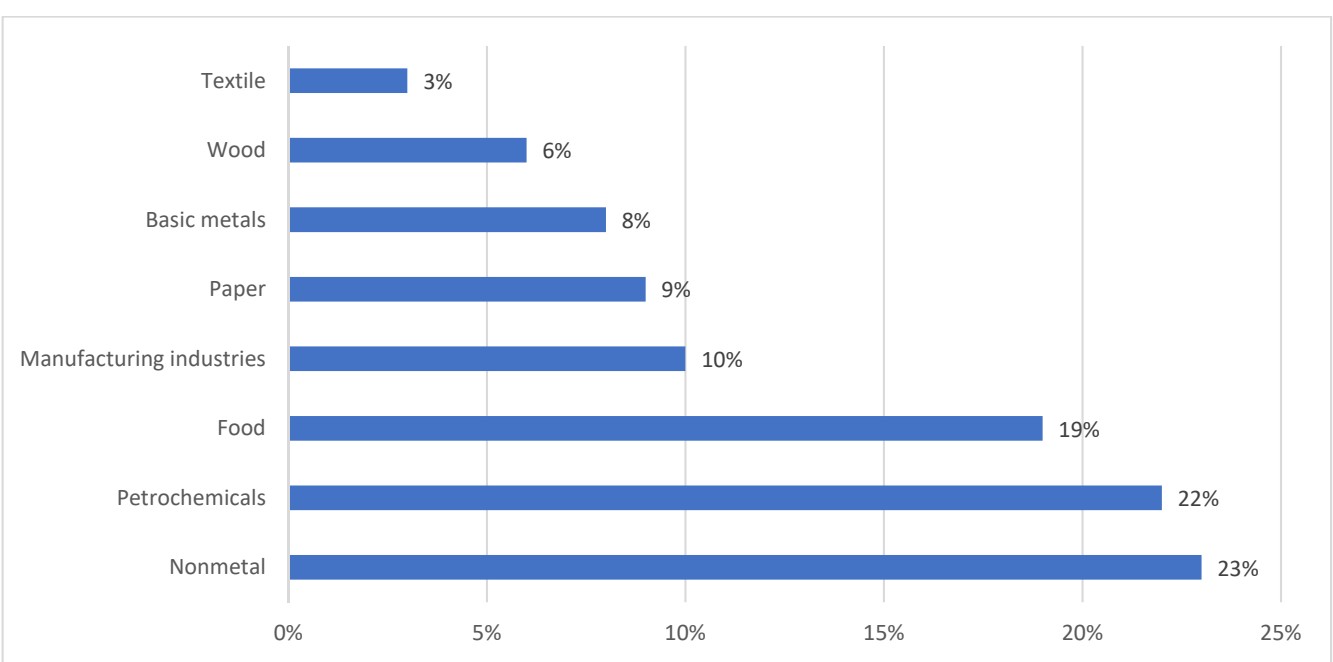

**Figure 3.** The major activity classifications of Bahrain's industrial sector (data sources: [121,122]).

## 5. Research Methodology

This study aims to verify the level of SDG implementation in the large industries in Bahrain based on the disclosure documents published on their websites. The methodology used to achieve this objective was based on analysis of annual and sustainability reports from the large industrial sector published between 2016 and 2020. The large industrial sector was selected because it is considered to be the major emitter of various pollutants (the major source of pollution) that create continuous pressures and impacts on the environment. Those pressures and impacts have magnified the significant need for the adoption of sustainability reports to enhance the capability of this sector to comply with the national environmental regulations and benefit from the implementation of SDGs. These reports were used for data and information collection as the annual reports are the most informative, public documents that companies publish on their websites. In addition, there is a positive indication towards the adoption of sustainability reports among this sector.

According to the first National Voluntary Review of the Kingdom of Bahrain on the 2030 Agenda for Sustainable Development, one of the challenges is how Bahrain maintains sustainable economic growth to provide an attractive environment for investment and better and diverse employment opportunities for citizens. Besides that, the new industrial policy in Bahrain concentrates on creating a profitable and environmental friendly sector that creates better economic prosperity and more jobs. Additionally, Bahrain is well positioned to focus on manufacturing opportunities that will benefit its economy and make maximum productive use of the scarce land, water and energy resources, while maintaining its high standards in environmental protection. In this regard, the 2030 Agenda stated that the industrial sector is considered one of the main stakeholders in achieving SDGs because it is the driver of economic growth. Therefore, there is a need to integrate the SDGs in the industrial sector policies to address the environmental problems resulting from this sector to achieve better environmental performance towards attaining sustainable industrial development. To do so, environmental reporting (e.g., GRI) is an efficient tool to measure the environmental commitments and performance of this sector. Figure 4 depicts the current research methodology.

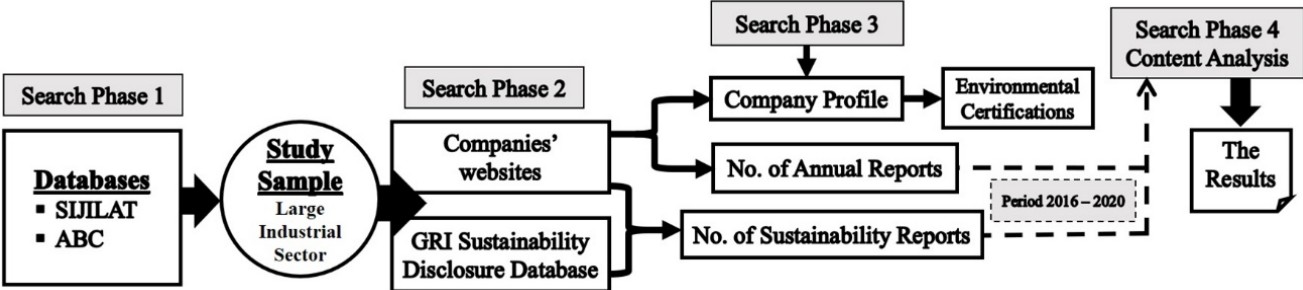

**Figure 4.** A depiction of the study research methodology.

*5.1. Sample and Data*

The samples included different large industrial firms that were registered in the Ministry of Industry, Commerce, and Tourism database (Business Licensing System (SIJILAT)), and included approximately 170 companies. In addition, the open database of the Arabian Business Community (ABC) Bahrain and companies' websites were used to collect the required information.

*5.2. Content Analysis of Annual and Sustainability Reports*

The content analysis method employed is a type of textual analysis [112], and it is a combination of qualitative and quantitative research approaches [125]. This methodology is usually employed in studying CSR and sustainability reports [126,127] to gain insights into the sustainable manufacturing field [128]. The study employed this method to assess the disclosed SDGs and the level of understanding of the SDG requirements. The quantitative content analysis method was then applied to identify the frequency of concepts, such as words or phrases, within the text. The study specified six keywords to use in search queries on the annual and sustainability reports (Table 2).

**Table 2.** Units (keywords) of the content analysis.

| Documents of Analysis | SDGs | Units of Analysis |
|---|---|---|
| Annual Reports Sustainability Reports | 8, 9, 11, 12, 13, 14, 15, and 17 | "SDG", "SDGs", "global goal", "sustainable", "sustainability", and "development" |

At the end of the selection process, only 11 sustainability reports from 3 companies were obtained, as depicted in Table 3.

**Table 3.** The number of sustainability reports published by year.

| Year | 2016 | 2017 | 2018 | 2019 | 2020 | Total |
|---|---|---|---|---|---|---|
| Number | 2 | 2 | 3 | 3 | 1 | 11 |

For the analysis purpose, 13 reporting analysis parameters (see Table 4) were obtained from the recommendations on SDG reporting provided by GRI, UN Global Compact, and the literature.

**Table 4.** Analysis parameters for evaluation of the sustainability reports.

| Analysis Parameters | Gulf Petrochemical Industries Company (GPIC) | | | | | Aluminum Bahrain (ALBA) | | | | | Bahrain Petroleum Company (BAPCO) * | | | | |
|---|---|---|---|---|---|---|---|---|---|---|---|---|---|---|---|
| | 2016 | 2017 | 2018 | 2019 | 2020 | 2016 | 2017 | 2018 | 2019 | 2020 | 2016 | 2017 | 2018 | 2019 | 2020 |
| Does the CEO's statement show a reference to the SDGs? | Y | Y | Y | Y | Y | N | N | Y | Y | | | | | Y | |
| Does the report present brief information about the SDGs? | Y | Y | Y | Y | Y | N | N | N | Y | | | | | N | |
| Does the report provide a reference relating the SDGs to the company? | Y | Y | Y | Y | Y | N | N | N | Y | | | | | N | |
| Does the report rank the SDGs as per priority? | Y | Y | Y | Y | Y | N | N | N | N | | | | | N | |
| Does the report refer to stakeholders? | Y | Y | Y | Y | Y | Y | Y | Y | Y | | | | | Y | |
| Does the report mention the method/tool utilized for disclosure of the SDGs? | Y | Y | Y | Y | Y | Y | Y | Y | Y | | | | | Y | |
| Does the report present the materiality assessment? | Y | Y | Y | Y | Y | Y | Y | Y | Y | | | | | Y | |
| Does the report link the company's environmental, economic, and social aspects with the SDGs? | Y | Y | Y | Y | Y | N | N | N | Y | | | | | N | |
| Does the report state that the company's KPIs are based on the SDGs? | Y | Y | Y | Y | Y | Y | Y | Y | Y | | | | | Y | |
| Does the report refer to the inclusion of sustainable development goals in the company's strategy/plan? | Y | Y | Y | Y | Y | N | Y | Y | Y | | | | | Y | |
| Does the report provide the outcomes/benefits of the implemented SDGs? | Y | Y | Y | Y | Y | N | N | N | Y | | | | | N | |
| Does the report mention the current projects/actions for achieving the SDGs? | Y | Y | Y | Y | Y | N | N | N | Y | | | | | Y | |
| Does the report mention the future projects/actions for achieving the SDGs? | Y | Y | Y | Y | Y | N | N | N | Y | | | | | Y | |

* The report for 2018–2019; Y: yes; N: no.

## 6. Results and Discussion

The study targeted large industrial companies, examining their annual and sustainability reports that were published on their websites. Based on an internet search, the results showed that 155 (91.2%) of the 170 companies had a website, 22 of the 155 websites not found, and 39 websites were of a poor standard, lacking information and data. The remaining 74 websites provided some useful information and data on local companies, and 20 websites were international, not containing or referencing any information or data about their branch in Bahrain.

The authors collected 29 (i.e., 17%) annual and sustainability reports from the firms' websites: 15 annual reports and 14 sustainability reports. From 15 annual reports, only 7 were focused on Bahrain, while 8 were international corporate reports without any mention of information and data about their branches in Bahrain. As far as sustainability reports are concerned, only three were focused on their environmental efforts and achievements in Bahrain, two were part of an annual report, and one was focused on the financial aspects. Therefore, this study focused on three companies to assess the status and contribution of their sustainability reports to the implementation of SDGs in Bahrain for the period 2016–2020.

Regarding the importance of environmental certifications, the quality, environment, health and safety, and CSR in Bahrain's industrial sector, refer to Section 0.1 (General) of ISO 9001:2015, which states that "the adoption of a quality management system is a strategic decision for an organization that can help to improve its overall performance and provide a sound basis for sustainable development initiatives" [129]. ISO 14000 International Standards series focuses mainly on the environmental pillar of sustainability and ISO 14001 is principally aimed at improving environmental performance and sustainability [130]. ISO 45001, which outlines a proactive approach, was launched in 2018 to replace the Occupational Health and Safety Management standard (OHSAS 18001:2007), aimed at assisting organizations to minimize their occupational health and safety risks by adopting efficient prevention and protection measures. Furthermore, CSR is considered to be a potent tool for strengthening synergies among stakeholders and reshaping their responsibilities, along with enhancing the private sector's role in attaining sustainable development. Moreover, there are thematic overlaps between CSR and SDGs that help to improve the model for sustainability and development growth [131]. A search of company certification profiles revealed that approximately 68 firms were certified with ISO 9001, while 39 were certified with ISO 14001, 31 firms were certified with OHSAS, and 17 with ISO 45001. Regarding CSR, 24 firms published their CSR on their websites. The results of this search do not represent the accuracy status of certification due to a lack of information, as mentioned above. However, the International Organization for Standardization (ISO) surveys recorded notable certification growth in Bahrain between 2000 and 2019 for both ISO 14001 and ISO 9001. Figure 5 shows a significant increase in the issuance of certificates since 2000. The number of certifications with ISO 9001 increased from 33 to 256 between the years 2000 and 2009, and increased to 435 (70%) in 2019. Concerning ISO 14001, the number of certificates that were issued increased from 42 to 141 during the period 2011–2012, and it reached 142 in 2019.

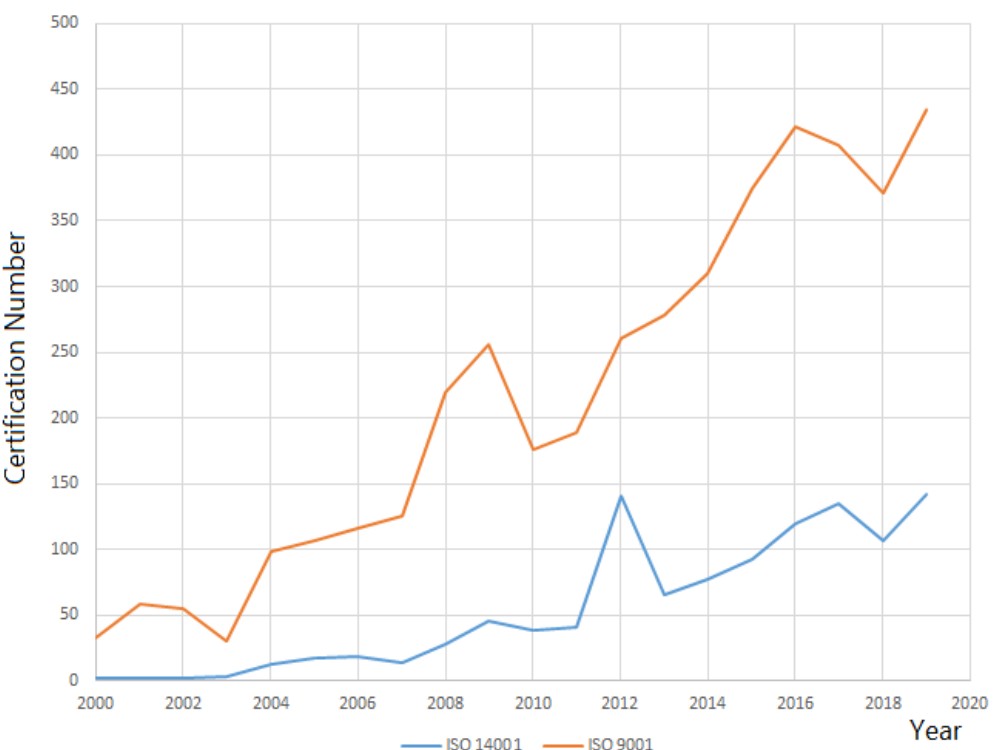

**Figure 5.** ISO 14001and ISO 9001 certification growth in Bahrain from 2000 to 2019 (data sources [129,132–136]).

Table 5 shows the number of SDG reports for the period 2016–2020 for three companies in Bahrain (Gulf Petrochemical Industries Company (GPIC), Aluminum Bahrain (ALBA), and Bahrain Petroleum Company (BAPCO)). Among all the sustainability reports, only one company referred to SDGs in the CEO's statement (GPIC). Since 2018, companies have started to include SDGs in their reports.

It was noted that the content and quality of the SDG reports were generally good and provided good information on SDGs, and only one company (GPIC) included all the SDGs in detail in its report for the five years. The reports of the other two companies lacked any references to relevant SDGs. Concerning prioritizing SDGs, only the GPIC report linked the priority areas to the SDGs, whereas the ALBA report indicated the priority areas without linking them to SDGs. It is worth mentioning that all the companies studied included their related stakeholders and materiality assessment in their reports. Furthermore, one company (GPIC) linked their environmental, economic, and social aspects with the SDGs in all reports, another company started doing so in 2019, while the third company did not. Concerning the inclusion of SDGs in the company's strategy or plan, only one company (GPIC) had included all SDGs in its strategic plan since 2016, another company included only SDG 12 and 13 in its plan, while the third included without mentioning SDGs in its plans. As shown in Table 2, the last three analysis parameters were presented in the reports of one company (GPIC), another company included them in its reports from 2019, while the third company provide brief information about last two parameters in its reports.

Importantly, according to the reports, all companies employed GRI standards as a method for sustainability reporting; therefore, it can be said that they implicitly refer to the SDGs indirectly. In this regard, the Global Reporting Initiative [137] published a valuable document (Linking the SDGs and the GRI Standards), which linked the 17 SDGs and the GRI standards, and the disclosures that pertain to each standard.

Figure 6 illustrates the materiality matrix compiled for the three companies (ALBA, BAPCO, and GPIC) based on their reports for 2019 and 2020. A report on the materiality matrix is optional for the companies, and the results clearly showed an increase in information on how a company identifies its related stakeholders and how its continuous reporting on

materiality reflects its seriousness in addressing significant aspects by complying with the GRI Standards. The matrix showed that environmental aspects (e.g., waste management and waste water) are highly significant, and most were adequately reported. In addition, the companies varied in rating the importance of environmental aspects because of their different business activities; for example, biodiversity has less impact on companies. For instance, in Jordan, environmental disclosure of companies is still at an early stage, and is related to pollution aspects followed by the financial information aspect related to the environment [97]. Therefore, it is significant to devote the integrated reporting of environmental information by requesting companies to disclose full coverage of all relevant environmental aspects [138].

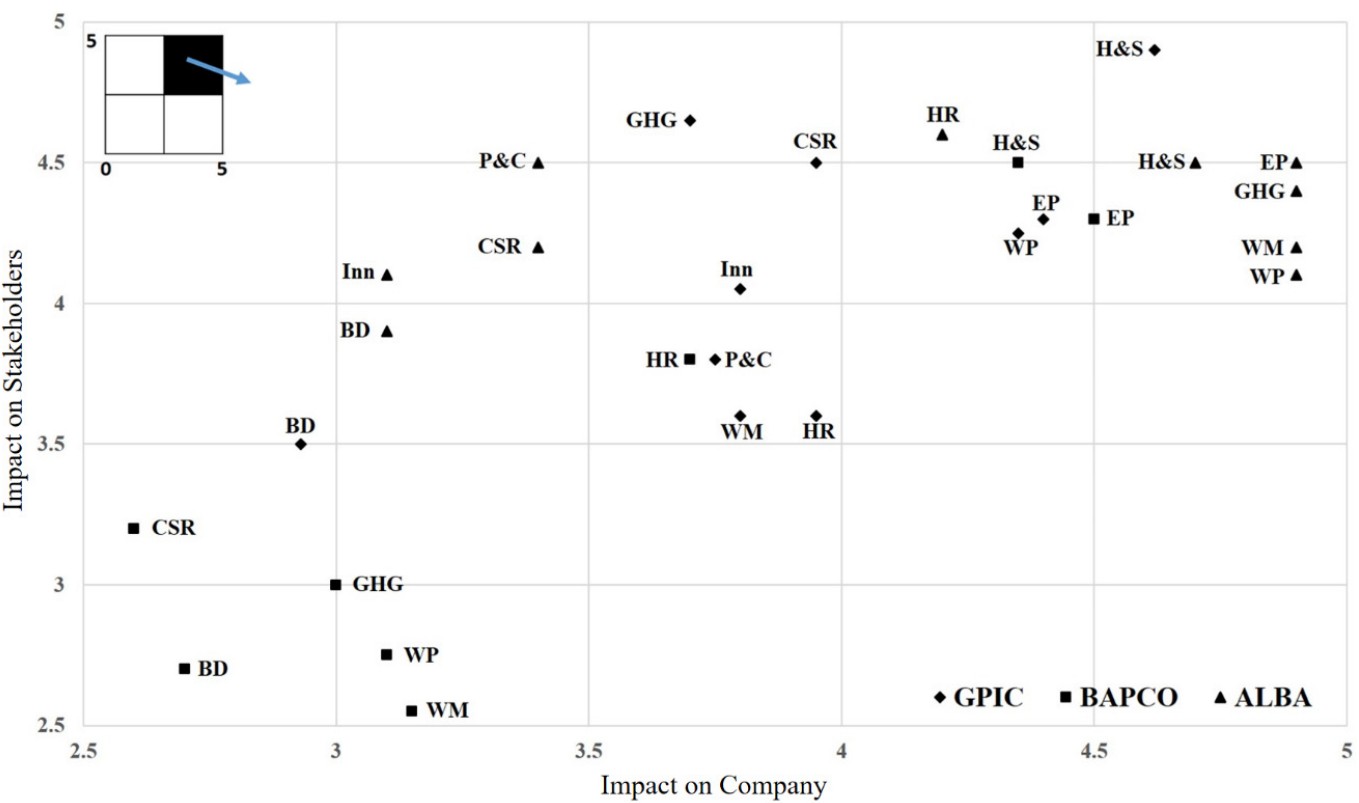

**Figure 6.** Sustainability materiality matrix compiled for the three companies (data sources [139–141]).

Figure 7 illustrates the mean of the results of the analysis of the three matrices, and each aspect of the matrix is labeled with the related SDGs. The results in the upper right-hand quarter of the figure show that greenhouse gases (GHS), waste management, and water pollution are the most significant sustainability aspects for the companies and their stakeholders. In addition, the matrix shows additional environmental aspects, such as health and safety, economic performance, and corporate social responsibility (CSR). Furthermore, it is interesting to note that the results show that the aspect of human rights is at the forefront of awareness by the industrial sector. The results indicate that the materiality matrix of sustainability reporting provides clear guidelines for the efficient allocation of resources, and therefore should be invested in as strategies and future plans to ensure the achievement of the expectations of stakeholders.

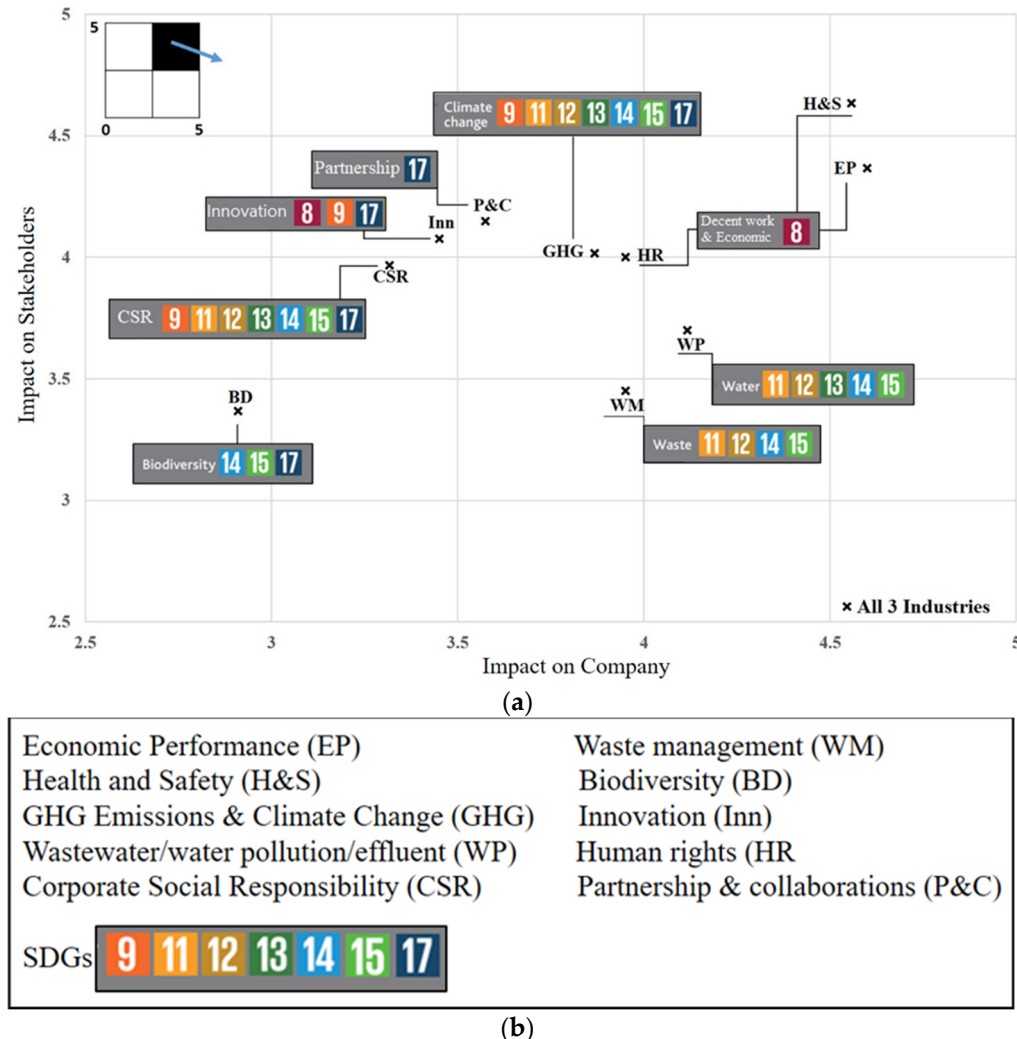

**Figure 7.** (**a**) Sustainability materiality matrix, (**b**) Key for figure.

Overall, the position of each sustainability aspect in the two matrices showed the consistency between the effectiveness of corporate sustainability reporting and the strategical importance of each aspect. Generally, this is a good indication, owing to the stakeholders' willingness to provide their views and make a judgment about the adequacy of company disclosures.

Figure 8 illustrates the ranking of the materiality aspects according to the combined materiality matrix. It shows that the companies pay great attention to economic performance (EP) and health and safety (H&S) aspects, whereas the stakeholders pay attention to water pollution (WP) and partnership and collaborations (P&C) aspects. While both similarly rank the health and safety (H&S) and human rights (HR) aspects, the greenhouse gases (GHG) aspect is ranked with minimal difference. In contrast, there is variance in the ranking of most other aspects, such as corporate social responsibility (CSR), waste management (WM), innovation (Inn), biodiversity (BD), and partnership and collaborations (P&C). It is clear that this study encourages companies to use materiality matrix analysis and to develop priority assessment measures to address and minimize the variance.

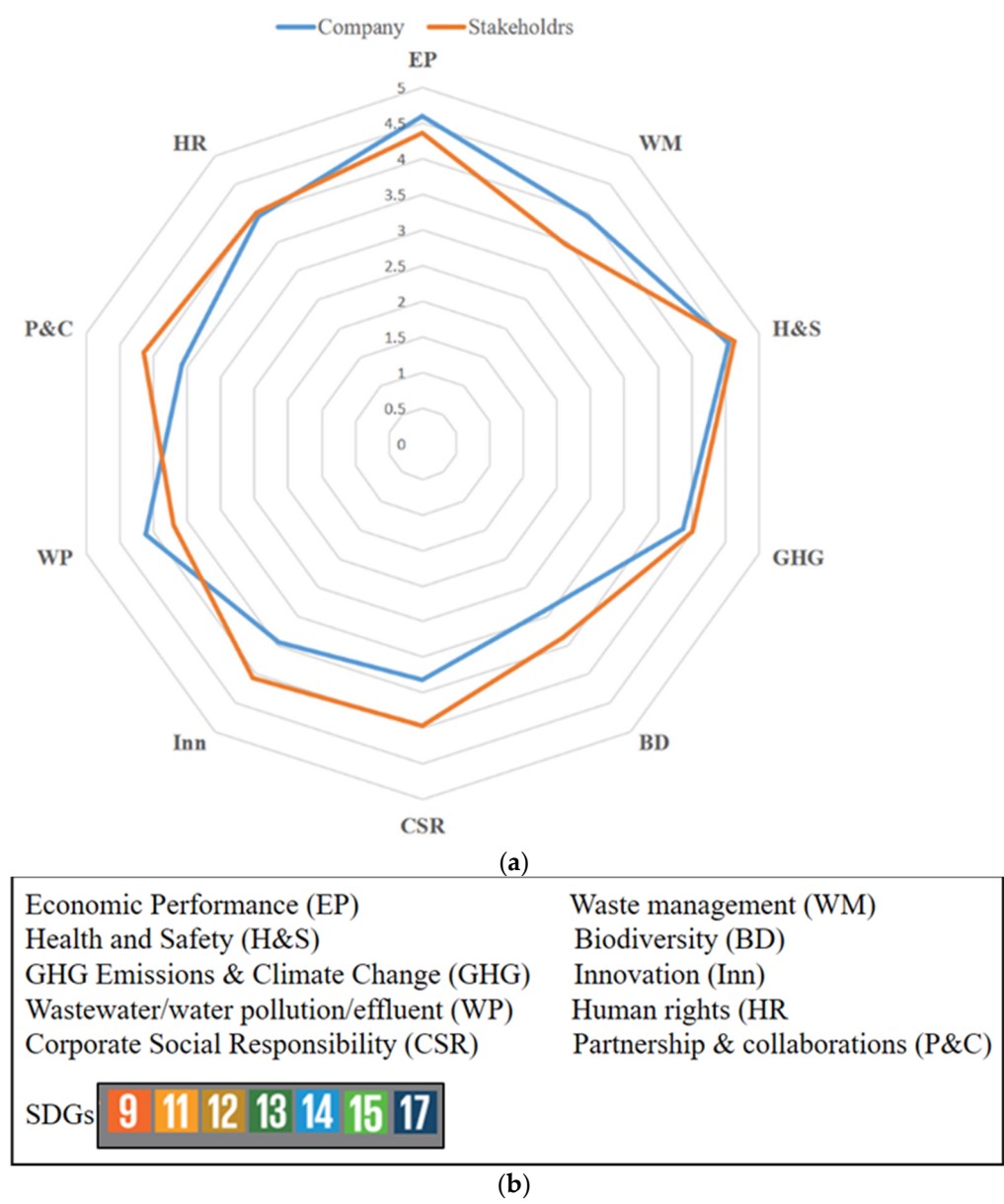

**Figure 8.** (**a**) Ranking of materiality according to the combined materiality matrix, (**b**) Key for figure.

Table 5 illustrates the priority of each SDG for each company based on the outcomes of the sustainability reports. Owing to the limited number of sustainability reports, it was difficult to evaluate the contribution of each goal individually, and there is a need for better data to conduct sustainability ratings for the industrial sector. Thus, at this stage, company interest is a positive indication to encourage other companies to start sustainability reporting.

**Table 5.** The priority of the sustainable development goals (SDGs) related to the industrial sector, according to the sustainability matrix compiled for the period 2019–2020.

| SDG | Brief Description | Priority (High, Medium, Low) | Achievements |
|---|---|---|---|
| 8 DECENT WORK AND ECONOMIC GROWTH | Promote sustained, inclusive, and sustainable economic growth, full and productive employment, and decent work for all | High | • GPIC has injected into the national economy since its inception USD 5.067 billion [139].<br>• BAPCO Modernization Program (BMP) has created 150 high-skilled permanents jobs [140].<br>• Aluminum sector represents about 12% of Bahrain GDP and it will increase to 15% with the expansion of the Line 6 project [141]. |
| 9 INDUSTRY, INNOVATION AND INFRASTRUCTURE | Build resilient infrastructure, promote inclusive and sustainable industrialization, and foster innovation | High | • GPIC won the Mohammed Rashid bin Maktoum Business Innovation Award in 2017 and 2019 [142].<br>• Bahrain Distributed Solar Energy Pilot has a future target to contribute to the national energy demand by 5% in 2025 and by 10% in 2035 [141]. |
| 11 SUSTAINABLE CITIES AND COMMUNITIES | Make cities and human settlements inclusive, safe, resilient, and sustainable | High | • The wasted heat generated from calciner plant is used to run a seawater desalination plant to produce potable water (+90%) for the local community [143]. |
| 12 RESPONSIBLE CONSUMPTION AND PRODUCTION | Ensure sustainable consumption and production patterns | High | • Advocating sustainability reporting as part of target 12.6 [144].<br>• Recycled 9197 MT of solid wastes, and hazardous waste has decreased by 5% [140].<br>• Since the launching of the recycling program in February 2017, approximately 123 MT of recyclable materials has been recycled [141]. |
| 13 CLIMATE ACTION | Take urgent action to combat climate change and its impacts | High | • Carbon dioxide recovery (CDR) plant captures about 450 MT/day of $CO_2$ from methanol reformer flue gases and recycles it [145].<br>• The intensity of GHG emissions has decreased by 3% [140].<br>• Solar energy consumption has increased to 12 TJ in 2019, compared with 10 TJ in 2018 [141]. |
| 14 LIFE BELOW WATER | Conserve and sustainably use the oceans, seas, and marine resources for sustainable development | High | • Strict compliance with national legal requirements for effluent discharge as the total of sewage effluent that was recycled and reused was 188,510 m$^3$ [140,146].<br>• Inauguration of the mangrove nursery was in 2018 [141]. |
| 15 LIFE ON LAND | Protect, restore, and promote sustainable use of terrestrial ecosystems, sustainably manage forests, combat desertification, halt and reverse land degradation, and halt biodiversity loss | High | • Several biodiversity projects were established, e.g., bird sanctuary, mangrove plantation, fish farm, and date palm trees [147].<br>• Establishment of Princess Sabeeka Oasis, which was divided into an aquatic area (lake), a landscaped area, and a vegetable garden [148].<br>• Establishment of Princess Sabeeka Park which contains around 51,000 plants, 560 palms, and 342 fruit and ornamental trees [141]. |
| 17 PARTNERSHIPS FOR THE GOALS | Strengthen the means of implementation and revitalize the global partnership for sustainable development | High | • Ongoing partnerships (e.g., UN Environment, Ministry of Education, InJaz, universities) [139].<br>• Ongoing partnerships (e.g., InJaz, Green School Award, universities) [141]. |

### 7. Conclusions and Recommendations

This research aims to examine the sustainability reports of companies by analyzing the status of sustainability aspects in their materiality matrices to assist in identifying and prioritizing the most significant sustainability issues for advancement in their future reporting and to improve their environmental performance. This study employs a content analysis.

In spite of the reporting concept being acknowledged in the 2030 Agenda and its SDGs, clarifying the role of the private sector in contributing to the implementation of SDGs is not an easy task due to the broad scope of the 17 goals, 169 targets, and more than 230 indicators that address an extensive range of prime challenges at national and global levels. Therefore, there is need for a considerable transformation of business models, plans, and strategies, specifically in terms of how this essential sector controls its operational impacts by creating balance among the sustainability pillars (society, environment, and the economy). After extensive investigation, this study reached a conclusion; among a large sector of Bahraini companies, "Bahrain Petroleum Company "BAPCO", "Gulf Petrochemical Industry Company "GPIC", and Aluminum Bahrain "ALBA" were found to have sustainability reports based on GRI guidelines.

The study of SDG reporting is a new research topic, despite the increased disclosure because of an awareness of the importance of SDGs. The results of this study cannot be generalized to the entire industrial sector in Bahrain because the data were limited to the five years from 2016 to 2020. The result of this study and analysis will assist in identifying and prioritizing the significant issues of sustainability to advance their future reporting for a better environmental performance, and to encourage other companies that did not establish sustainability reports to follow suit. However, the study opens a window to further studies on different business sectors and different types of corporate reports, to investigate the possible consequences of SDG disclosures in terms of environmental, social, and economic impact.

Mentioning SDGs in company reports is an opportunity to measure the implementation of SDGs and to strengthen their disclosure. Despite the limited number of SDG disclosures, it is necessary to study the potential impact of SDG disclosures on corporate performance, and to critically identify the main related stakeholders who are more influential in increasing the number of these disclosures. The results showed that companies have started sustainability reporting; publishing reports on their websites reflect company responsibility towards the environment and society. The subsequent sustainability reports showed tangible changes in the companies in terms of the quality of data presented in the reports, strategies, the role of related stakeholders, and reputation. Furthermore, this study focused on analyzing the sustainability reports to determine whether those companies have contributed to implementation of SDGs goals, targets, and indicators.

Materiality analysis is an essential part of both sustainability and integrated reporting, and it plays an important role in the disclosures. Therefore, a sustainability materiality matrix is used to identify significant related environmental aspects, and outline the report content. The outcomes of the materiality matrix analysis in this study have confirmed the crucial role that the industrial sector plays in the implementation of SDGs. Thus, instead of having short-term goals of limited benefit, SDGs have widened the scope for multiple future expectations concerning what should be developed and sustained, and for how long. Furthermore, the materiality matrix analysis of the reports of all companies revealed a focus on sustainability-related aspects and commitments to stakeholder engagement. The purpose of stakeholder participation in a materiality matrix is to push companies to provide more transparency and better information in their reports. This study recommends that companies use materiality analysis as a tool for reporting to reflect their commitment and to improve the report content by utilizing GRI guidelines.

## 8. Recommendations

1. The results of this study showed that SDGs are the key to improving corporate reporting to incorporate all environmental aspects, and may lead to a shift to the adoption of sustainability reports instead of annual reports. In addition, building an awareness of the importance of SDGs results in companies becoming more sustainability-oriented. According to Lozano et al., to make companies more sustainability-oriented, it is essential to grasp the comprehensive characteristics of corporate reporting and the synergies among technical, management, and organizational operations across time [86].

2. It is time to adopt sustainability reporting into law. The aim of this article was to study sustainability reports to highlight their importance in enhancing the positive aspects of companies and to bring out the benefits of integrating SDGs into the reports. Banik and Lin stated that enforcement of the regulations on sustainable development is still a major challenge [149]. According to Gibbon and Joshi (1999) that, although there is an absence of environmental accounting for external reporting purposes, Bahrain's companies are aware that environmental issues will affect their practice in the future. Further, this study revealed that about 50% of companies have environmental reporting for internal purposes, and more than 40% believe that their companies will start environmental reporting by 2000 [150]. In this context, Pizzi et al. stated that the presence of SDGs could be beneficial for differentiating compliance reports and identifying the reports aiming to impact sustainable development [49].

3. To prevent the embellishment of a company's positive performance or reputation in order to raise the credibility and authenticity of its reporting, this study recommends proposing and placing a list of the main environmental material aspects for each industrial sector based on local environmental laws and regulations, to ensure that the sustainability reports reflect the actual state of the environment.

4. Due to a lack of technical or vocational knowledge on particular aspects of company performance for some stakeholders, increasing the reporting efficacy and quality of the information in sustainability reports is recommended. In this vein, there is a need to increase the capacity-building of the stakeholders through training and consultation meetings, including effective coordination to agree on a resilience methodology of reporting.

5. It is essential to conduct a continuous evaluation of the company's sustainability efforts that enables the company to establish strategies to overcome resistance to institutional improvements and barriers to change.

6. Since the companies that responded to self-sustainability reporting integrated the national legislation compliance requirements with sustainability reporting, this has given policymakers the opportunity to revise or amend future legislation to ensure compliance with the law through sustainability reporting activities. In this regard, Rashed stated that, at the national level, there is an urgent need for serious action to activate the vision of the 2030 Agenda and its core 17 SDGs by promoting policy coherence by amalgamating the legal policy aspects of the environmental, economical, technical, institutional, and sociocultural dimensions [151].

7. The document "Linking the SDGs and the GRI Standards" assists and facilitates reporting by organizations by employing GRI standards to evaluate how they impact and comply with the SDGs. Therefore, companies should seize this opportunity to modify and improve their sustainability reports for the coming years.

8. Companies should invest in improvements through new initiatives targeting their stakeholders to enhance efficient cooperation and coordination on various matters, and enable identification of those material aspects that are not adequately disclosed.

9. In this regard, the government should take the opportunity to build joint sustainable initiatives to make room for the industrial sector to play its essential part as a key stakeholder at the national level. It is also worth highlighting some companies' achievements regarding each SDG.

**Author Contributions:** A.H.R.: As this manuscript is part of my thesis, I prepared the manuscript under the direction of supervisors. S.A.R.: As a co-author, I contributed in the advice and supervision of the PhD student as a main supervisor and contributed with the co-supervisor in setting the main concepts of the manuscript; I also reviewed and edited the drafts of the manuscript. A.Y.A.-M.: As a co-author, I contributed to the conceptualization of the setting of the manuscript; I reviewed and edited the drafts of the manuscript. As a co-supervisor, I contributed to the advice and supervision of the PhD student with the main supervisor. All authors have read and agreed to the published version of the manuscript.

**Funding:** This research received no external funding.

**Institutional Review Board Statement:** Not applicable.

**Informed Consent Statement:** Not applicable.

**Data Availability Statement:** Not applicable.

**Conflicts of Interest:** The authors declare no conflict of interest.

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
