# Peer review of "Towards Effective Environmental Sustainability Reporting in the Large Industrial Sector of Bahrain"

_sustainability, doi:10.3390/su14010219_

Round 1
Reviewer 1 Report
The paper deals with a very interesting topic in a context such as that of Bahrain which is still little explored by academic literature. It is very well built and is easy to read and understand.
I suggest that the authors only include a few more recent articles on the topic of environmental reporting. In this regard, I suggest you include the following articles:
Gerged, A. M. (2021). Factors affecting corporate environmental disclosure in emerging markets: The role of corporate governance structures. Business Strategy and the Environment, 30 (1), 609-629.
García-Sánchez, I. M., Raimo, N., & Vitolla, F. (2021). Are Environmentally Innovative Companies Inclined towards Integrated Environmental Disclosure Policies ?. Administrative Sciences, 11 (1), 29.
Gerged, A. M., Beddewela, E., & Cowton, C. J. (2021). Is corporate environmental disclosure associated with firm value? A multicountry study of Gulf Cooperation Council firms. Business Strategy and the Environment, 30 (1), 185-203.
Raimo, N., de Nuccio, E., & Vitolla, F. (2021). Corporate governance and environmental disclosure through integrated reporting. Measuring Business Excellence.
Good luck.
Author Response
the proposed references has been added.
Suggest articles |
Actions |
Gerged, A. M. (2021). Factors affecting corporate environmental disclosure in emerging markets: The role of corporate governance structures. Business Strategy and the Environment, 30(1), 609-629. |
Reference no.97 line 192-193, page 6 line 408-410, page 15 |
García-Sánchez, I. M., Raimo, N., & Vitolla, F. (2021). Are Environmentally Innovative Companies Inclined towards Integrated Environmental Disclosure Policies?. Administrative Sciences, 11(1), 29. |
Reference no.94 line 197-181, page 6 line 191-192, page 6 |
Gerged, A. M., Beddewela, E., & Cowton, C. J. (2021). Is corporate environmental disclosure associated with firm value? A multicountry study of Gulf Cooperation Council firms. Business Strategy and the Environment, 30(1), 185-203. |
Reference no.95
line 181-184, page 6 |
Raimo, N., de Nuccio, E., & Vitolla, F. (2021). Corporate governance and environmental disclosure through integrated reporting. Measuring Business Excellence. |
Reference no.138
line 410-412, page 15 |
Reviewer 2 Report
This is an interesting study. Here are some of my comments:
- What is the main question addressed by the research?
Firstly, the aim of the study is to research on the sustainability reports of several companies by analysing the status of sustainability aspects in the company matrices to assist in identifying and prioritising the significant sustainability issues, for advancing the future reporting and better environmental performance. While the aim of study is fairly clear, the problem statement or main question is not very clear within the abstract and introduction. There are bits and pieces of the research gaps in the introduction, but these aspects need to be refined into a proper problem statement. I suggest that the authors allocate a specific paragraph in the introduction just before outlining the aim of study to address the problem statement or main question, and prepare a one-sentence problem statement for the abstract as well, also before the aim of study is outlined.
- Do you consider the topic original or relevant in the field, and if so, why?
Yes. Generally, it is not that sustainability reporting does not exist, but the originality in this study concerns the way some tools are used to enable and improve sustainability reporting, with an emphasis on the industrial sector in Bahrain. The relevance of the study is also satisfactory since it is in line with the SDGs.
- What does it add to the subject area compared with other published material?
The article does highlight a number of comparisons to company reports, which is an effective way to observe its viability. However, it would be good to add a few more comparisons with findings from peer reviewed journal articles, just to see where this study stands in the academic community of Sustainability.
- What specific improvements could the authors consider regarding the methodology?
It is important to highlight the reason as to why Bahrain was the focus of the authors in this study. The authors should also describe how the application of this study can be extended to other sectors or countries.
- Are the conclusions consistent with the evidence and arguments presented and do they address the main question posed?
The conclusions are satisfactory, but we urge the authors to tie the conclusion with the addressal of the problem statement or aim of study from item 1.
- Are the references appropriate?
There references are appropriate and very diverged in terms of types of sources. While the list of references is comprehensive enough, perhaps the authors could add just a few more (3-6 more) very recent (year 2021) peer reviewed journal references that are very relevant to the study (in line with item 3).
- Please include any additional comments on the tables and figures.
The tables and figures are fine. I encourage the authors to officially get their manuscript proofread before the actual publication.
Thank you, well done, and all the best.
Author Response
# |
Comments |
Actions |
1 |
What is the main question addressed by the research? Firstly, the aim of the study is to research on the sustainability reports of several companies by analysing the status of sustainability aspects in the company matrices to assist in identifying and prioritising the significant sustainability issues, for advancing the future reporting and better environmental performance. While the aim of study is fairly clear, the problem statement or main question is not very clear within the abstract and introduction. There are bits and pieces of the research gaps in the introduction, but these aspects need to be refined into a proper problem statement. I suggest that the authors allocate a specific paragraph in the introduction just before outlining the aim of study to address the problem statement or main question, and prepare a one-sentence problem statement for the abstract as well, also before the aim of study is outlined. |
line 13-15, page 1 line 99-101, page 3
|
2 |
Do you consider the topic original or relevant in the field, and if so, why? Yes. Generally, it is not that sustainability reporting does not exist, but the originality in this study concerns the way some tools are used to enable and improve sustainability reporting, with an emphasis on the industrial sector in Bahrain. The relevance of the study is also satisfactory since it is in line with the SDGs. |
No comments |
3 |
What does it add to the subject area compared with other published material? The article does highlight a number of comparisons to company reports, which is an effective way to observe its viability. However, it would be good to add a few more comparisons with findings from peer reviewed journal articles, just to see where this study stands in the academic community of Sustainability. |
line 181-184, page 6 line 408-410, page 15 |
4 |
What specific improvements could the authors consider regarding the methodology? It is important to highlight the reason as to why Bahrain was the focus of the authors in this study. The authors should also describe how the application of this study can be extended to other sectors or countries. |
line 466-469, page 17 |
5 |
Are the conclusions consistent with the evidence and arguments presented and do they address the main question posed? The conclusions are satisfactory, but we urge the authors to tie the conclusion with the addressal of the problem statement or aim of study from item 1. |
line 463-466, page 17 |
6 |
Are the references appropriate? There references are appropriate and very diverged in terms of types of sources. While the list of references is comprehensive enough, perhaps the authors could add just a few more (3-6 more) very recent (year 2021) peer reviewed journal references that are very relevant to the study (in line with item 3). |
Five recent references has been added, numbers 94, 95, 97, 138 and 149 |
7 |
Please include any additional comments on the tables and figures. The tables and figures are fine. I encourage the authors to officially get their manuscript proofread before the actual publication. |
MDPI confirms that it has received payment of English editing (invoice english-38025) |
Round 2
Reviewer 2 Report
After going through the revision, I found that there are still comments that have not been explicitly addressed. Perhaps the reviewer will try to be more specific in case the authors misunderstood:
1) What specific improvements could the authors consider regarding the methodology?
Round 1 (original) comment: It is important to highlight the reason why Bahrain was the focus of the authors in this study.
Round 2 (present) comment: Basically, the reviewer meant that the authors should explicitly state why they chose Bahrain as a centre point or emphasis in this particular study. Was there a particular reason for selecting Bahrain for this study? For instance, was the reason due to a specific performance indicator among the large industries in Bahrain? Please clarify this point in the methodology, or make it obvious for the reader.
2) Are the conclusions consistent with the evidence and arguments presented and do they address the main question posed?
Round 1 (original) comment: We urge the authors to tie the conclusion with the addressal of the problem statement or aim of the study.
Round 2 (present) comment: The reviewer would actually be interested in seeing how the authors close the loop of this study within the conclusion by addressing the problem statement. In this case, the authors have chosen to use research questions (which are a good idea) to break down the problem statement: "RQ1 - Do Bahraini companies have sustainability reports and if so, are these reports based on GRI guidelines?" and "RQ2 - Has the status of their sustainability reports been analyzed?". So, the reviewer hopes that the authors can specifically and systematically address these two questions in the conclusion to close the loop of the study's findings. This will make the conclusion more clear cut and solid.
The rest of the study is good, and all other comments have been satisfied. Well done. Thank you.
Author Response
Comments |
Actions |
1) What specific improvements could the authors consider regarding the methodology? Round 1 (original) comment: It is important to highlight the reason why Bahrain was the focus of the authors in this study. Round 2 (present) comment: Basically, the reviewer meant that the authors should explicitly state why they chose Bahrain as a centre point or emphasis in this particular study. Was there a particular reason for selecting Bahrain for this study? For instance, was the reason due to a specific performance indicator among the large industries in Bahrain? Please clarify this point in the methodology, or make it obvious for the reader. |
Page 10 Line 295 - 309 |
2) Are the conclusions consistent with the evidence and arguments presented and do they address the main question posed? Round 1 (original) comment: We urge the authors to tie the conclusion with the addressal of the problem statement or aim of the study. Round 2 (present) comment: The reviewer would actually be interested in seeing how the authors close the loop of this study within the conclusion by addressing the problem statement. In this case, the authors have chosen to use research questions (which are a good idea) to break down the problem statement: "RQ1 - Do Bahraini companies have sustainability reports and if so, are these reports based on GRI guidelines?" and "RQ2 - Has the status of their sustainability reports been analyzed?". So, the reviewer hopes that the authors can specifically and systematically address these two questions in the conclusion to close the loop of the study's findings. This will make the conclusion more clear cut and solid. |
Page 17 Line 471 - 475 Line 483 – 486 Line 505 – 508 |
Addition to recommendation no. 2 |
Page 18 Line 535 - 541 |
